# Effect of Heating on Hot Deformation and Microstructural Evolution of Ti-6Al-4V Titanium Alloy

**DOI:** 10.3390/ma16020810

**Published:** 2023-01-13

**Authors:** Dechong Li, Haihui Zhu, Shuguang Qu, Jiatian Lin, Ming Ming, Guoqing Chen, Kailun Zheng, Xiaochuan Liu

**Affiliations:** 1School of Materials Science and Engineering, Dalian University of Technology, Dalian 116024, China; 2School of Mechanical Engineering, Dalian University of Technology, Dalian 116024, China; 3School of Mechanical Engineering, Xi’an Jiaotong University, Xi’an 710049, China

**Keywords:** dual-phase titanium alloy, heating rate, microstructure, phase model, hot forming

## Abstract

This paper presents a systematic study of heating effects on the hot deformation and microstructure of dual-phase titanium alloy Ti-6Al-4V (TC4) under hot forming conditions. Firstly, hot flow behaviors of TC4 were characterized by conducting tensile tests at different heating temperatures ranging from 850 °C to 950 °C and heating rates ranging from 1 to 100 °C/s. Microstructure analysis, including phase and grain size, was carried out under the different heating conditions using SEM and EBSD. The results showed that when the heating temperature was lower than 900 °C, a lower heating rate could promote a larger degree of phase transformation from α to β, thus reducing the flow stress and improving the ductility. When the temperature reached 950 °C, a large heating rate effectively inhibited the grain growth and enhanced the formability. Subsequently, according to the mechanism of phase transformation during heating, a phenomenological phase model was established to predict the evolution of the phase volume fraction at different heating parameters with an error of 5.17%. Finally, a specific resistance heating device incorporated with an air-cooling set-up was designed and manufactured to deform TC4 at different heating parameters to determine its post-form strength. Particularly, the yield strength at the temperature range from 800 °C to 900 °C and the heating rate range from 30 to 100 °C/s were obtained. The results showed that the yield strength generally increased with the increase of heating temperature and the decrease of heating rate, which was believed to be dominated by the phase transformation.

## 1. Introduction

Dual-phase titanium alloys have been widely used in the aerospace industry for manufacturing complex-shaped thin-walled components [1,2] due to their high strength-to-weight ratio, excellent corrosion resistance and high heat resistance [3]. Titanium alloys have strong deformation resistance and severe spring-back drawback that forms at room temperature, which may damage forming tools and reduce the accuracy of formed components [4]. In order to overcome the low formability at room temperature, superplastic forming (SPF) [5], hot stamping [6] and hot metal gas forming (HMGF) [7] were developed for forming complex-shaped thin-walled parts in recent years, featuring low deformation resistance and high forming precision [8,9]. Heating to elevated temperatures is the first and key step in hot forming processes. The microstructure, such as phase and grain size, varies with the forming temperature [10,11], which determines the subsequent hot flow behavior and post-form mechanical properties [12]. To facilitate the design of reasonable heating variables, particularly heating rate and temperature, a thorough understanding of the correlation between micro and macro-properties, i.e., microstructure evolution and hot flow, at various heating parameters is of vital importance to the hot forming of TC4 sheets.

The relationship between the mechanical properties, phase transformation and grain size evolution of titanium alloys at different heating temperatures has been studied [13,14]. Compared to the extensive research on the forming temperature, studies on the heating rate effect on the hot forming of titanium alloy sheets are limited to date. Xiao et al. [15] conducted uniaxial tensile tests on TC4 sheets in the temperature range of 923–1023 K. The real stress-strain curves showed that the flow stress decreased with the increase of temperature and the decrease of strain rate, but the effect of heating rate on the mechanical properties before and after forming was not considered. Wang et al. [16] studied both β phase transformation during rapid heating and martensitic phase transformation during cooling through high-speed machining of TC4. A stress-temperature-induced phase transformation model based on Avrami and Clausius-Clapeyron equations was proposed to predict the phase transformation from α to β during the rapid heating process. However, the mechanism of phase transformation at specific heating rates was not discussed in this study. Elmer et al. [17] conducted in-situ X-ray diffraction experiments to observe the phase transformation of TC4 alloys during heating using synchrotron radiation. It was observed that the diffusion of the β stable element V was inhibited as the heating rate increased, which led to a decrease in the β phase transformation temperature. However, the study was mainly limited to the phase transformation kinetics without investigating the phase transformation effect on the high-temperature deformation and the post-form properties. In addition, Surya et al. [18,19] studied the effects of input factors such as speed, feed and depth of cut on the material removal rate and the surface roughness of TC4 titanium alloy, and established a high-precision mathematical prediction model to determine the optimal parameter combination for cutting performance. This modeling idea is also applicable to the high-temperature deformation of TC4. It is, therefore, necessary to build a material model based on temperature and microstructure and to formulate a reasonable forming process by predicting the phase distribution of materials at high temperatures, in order to comprehensively understand the subsequent heat flow and strength variations after forming [20].

To quantify the heating rate, the conventional environmental heating furnace has some limitations, such as uncontrollable heating rate and low thermal efficiency. In order to accurately control the temperature change rate during the heating process, various heating methods were studied. Mori et al. [21] and Maeno et al. [22] applied a resistance heating method to the hot stamping of high-strength steel and dual-phase titanium alloy parts. Due to a high heating rate being used, negligible oxidation occurred on the sheet surface, and the current passed directly through the stamping parts, thus improving the heating efficiency. Therefore, the resistance heating technology is suitable for adjusting the current output and controlling the temperature change during the heating process [23]. Wang et al. [24] used resistance heating to heat the workpiece to 850–950 °C at heating rates of 4 °C/s and 100 °C/s, respectively, then cooled it to 700 °C to simulate the workpiece heating to the target temperature at different heating rates, and finally transferred it to the die for the hot stamping process, in order to investigate the effect of the heating rate on material ductility and post-form strength. However, the results of this study were not applicable to isothermal forming processes at specific temperatures without transfer processes, and little research has been performed in this field.

In this study, the heating effect on the microstructure evolution and hot flow of TC4 was systematically studied using the resistance heating method. The hot flow and phase evolution at different heating parameters were characterized by using a Gleeble thermal-mechanical simulator (Gleeble, New York, NY, USA), and the post-form strength was compared using the specimens that were heat treated using a self-developed resistance heating device. Additionally, a phenomenological phase transformation model was established based on the phase transformation kinetics during the heating process. The performed research is able to provide important guidance for the design of heating parameters, i.e., heating rate and temperature, for the various hot forming processes of TC4 sheets.

## 2. Experimentation

### 2.1. Material

Dual-phase titanium alloy TC4 sheets with a thickness of 1.5 mm were used in this study. The chemical composition is shown in Table 1, and the microstructure is shown in Figure 1, in which the white region represents the β phase and the dark region represents the α phase. The initial microstructure consisted of an equiaxed α phase with a volume fraction of 87% and a fine β phase with a volume fraction of 13%. The initial heat treatment temper and annealing condition had a yield strength and ultimate tensile strength of 1056 MPa and 1071 MPa, respectively. In addition, the elongation of TC4 was 18.5%.

### 2.2. Resistance Heating Tests

#### 2.2.1. Gleeble Hot Tensile Tests and Microstructure Characterization

In order to examine the effect of the heating process on the hot flow behaviors and microstructure evolution of TC4, Gleeble hot tensile tests were performed at various heating temperatures and heating rates using a Gleeble 3800 thermal-mechanical simulator, which enabled the maximum heating rate to reach 10,000 °C/s and allowed for precise control of temperature within ±1 °C.

Figure 2a shows the experimental flow chart of the high-temperature thermodynamic testing. Regarding the Gleeble hot tensile tests, conditions of different heating rates, temperatures and strain rates were performed as follows. The heating rates of 1 °C/s and 10 °C/s were selected, enabling the rate magnitudes of practical processes to be covered. Three temperatures of 850 °C, 900 °C and 950 °C were used at each heating rate. Once the specimen reached the target temperature, hot tensile tests at a strain rate of 0.1/s were conducted. The high-temperature tensile properties of TC4 at different heating rates and heating temperatures were then measured. The temperature was continuously monitored by the thermocouple welded at the center of the specimen to precisely feedback-control the heating rates and heating temperatures during testing. In the experimental group of heat treatments used to characterize the microstructure of TC4 during heating, the same heating history as that of the tensile tests at high temperatures was used. The specimen was immediately water-quenched after a soaking time of 1 s to maintain the microstructure. Microstructure observations, i.e., SEM and EBSD, were used to characterize the phase transformation and grain evolution of the specimens, and the quantitative microstructure results with statistical significance were subsequently obtained.

Figure 2b shows the specimen size and shape for the tensile and heat treatment tests. The dog-bone-shaped specimen was used with a length of 60 mm, a width of 12 mm and a thickness of 1.5 mm, which was machined from the as-received sheet using Electrical Discharge Machining along the rolling direction. The length of the parallel zone of the specimen could guarantee a homogeneous temperature zone of 10 mm in length in the middle. Therefore, secondary sampling for EBSD and SEM was carried out in the center of the sample to ensure the uniformity of its temperature history.

#### 2.2.2. Post-Form Strength Characterization

In order to investigate the rapid heating of TC4 sheets as well as the in-line cooling to characterize the post-form strength at different heating parameters, a resistance heating experimental device for titanium alloy sheets was designed and manufactured as schematically shown in Figure 3.

The current in the device was output by the high-frequency switching power supply, flowing directly from the copper ribbon to the copper electrode and passing through the TC4 sheet. Al_2_O_3_ ceramics were bolted between the copper electrode and the upper/lower plates to facilitate the insulation, ensuring that the current flew to the TC4 sheet completely through the copper electrode.

A plurality of gas channels was positioned inside the upper and lower plates, and connected with an adjustable gas source to provide pressure up to 0.8 MPa. Meanwhile, the outlet of each gas channel was connected with gas nozzles that were evenly distributed on both sides of the TC4 sheet, thus achieving uniform cooling of the sheet. The flowmeter was used to monitor the gas flow in real-time so as to control the cooling rate of the TC4 sheet. The k-type thermocouple was welded on the surface of the sheet, and the other end was inserted into the temperature recorder to facilitate the real-time monitoring and recording of the temperature.

Therefore, the device was able to perform the static treatment experiments at different heating rates, heating temperatures and cooling conditions to simulate the industrial forming process of titanium alloys.

Figure 4 illustrates the temperature histories of the resistance heating experiments. Three heating rates of 30 °C/s, 60 °C/s and 100 °C/s were selected. At each heating rate, three temperatures of 800 °C, 850 °C and 900 °C were used. Once the sheet reached the target temperature, the sheet immediately cooled to room temperature at a cooling rate of 200 °C/s using the gas flow, thus restoring the microstructure after heating. The temperature was continuously monitored by the thermocouple welded at the center of the sheet to precisely monitor the heating rates, cooling rates and heating temperatures. Sub-sized dog-bone-shaped specimens were designed and machined (shown in Figure 4) from the sheets to measure the strength and investigate the effect of temperature on the mechanical properties of TC4.

## 3. Results and Discussion

### 3.1. Flow Behaviors of TC4 at Different Forming Temperatures and Heating Rates

The effects of forming temperature and heating rate on the flow behavior of TC4 were investigated based on the Gleeble tensile tests and microstructure results. The true stress-strain curves shown in Figure 5 indicated that the maximum flow stress and elongation of TC4 decreased with the increase of the forming temperature. Specifically, when the heating rate was 1 °C/s, the maximum flow stress was 230 MPa, 149 MPa and 96 MPa at the forming temperatures of 850 °C, 900 °C and 950 °C, respectively, suggesting a 58% reduction with an increase in temperature from 850 °C to 950 °C, as shown in Figure 6a. Similarly, the maximum flow stress of TC4 decreased by 53% from 242 MPa to 114 MPa with the increase in forming temperature from 850 °C to 950 °C at a heating rate of 10 °C/s. In addition, the maximum flow stress at the heating rate of 10 °C/s was higher than that at the heating rate of 1 °C/s. Notably, when the heating temperature was 950 °C, the maximum flow stress increased by 19% from 96 MPa to 114 MPa with the increase in heating rate from 1 °C/s to 10 °C/s.

It was found that the elongation of TC4 decreased with an increase in forming temperature as well, as shown in Figure 6b. Specifically, the elongation decreased from 58% at a forming temperature of 850 °C to 43% at a forming temperature of 950 °C when a heating rate of 1 °C/s was applied. However, the effect of forming temperature on the elongation was negligible when the heating rate increased to 10 °C/s. When the heating temperature was 950 °C, the elongation was increased by 14% at a heating rate of 10 °C/s, compared with that at a heating rate of 1 °C/s.

### 3.2. Effects of Forming Temperature and Heating Rate on the Phase Transformation

The microstructural phase distributions of TC4 were obtained while investigating the effects of different forming temperatures and heating rates on the maximum flow stress and elongation of TC4, as shown in Figure 7 and Figure 8, in which the white region represents the β phase and the dark region represents the α phase. The β phase is a high-temperature stable phase, which gradually changes from the α phase during the heating process and mainly depends on the temperature and heating rate of the process. It could be seen that the volume fraction of the β phase increased with the increase in heating temperature when the heating rate was constant and decreased with the increase in heating rate when the heating temperature was constant. Moreover, the β phase microstructure was found to be finer at a lower heating temperature and a higher heating rate.

In order to investigate the effect of the β phase volume fraction on the mechanical properties clearly, a statistical analysis of the β phase volume fraction was conducted, as shown in Figure 9a, which was found to be 34%, 47% and 72% at the forming temperatures of 850 °C, 900 °C and 950 °C, respectively, when a heating rate of 1 °C/s was applied. It had been reported [25] that the α phase with a hexagonal compact packing structure was approximately three times stronger than the β phase with a body-centered cubic structure at an elevated temperature. Meanwhile, the dynamic recrystallization of TC4 increased with the increase in forming temperature, thus softening the material and reducing its deformation resistance. As a combined result of the dynamic recrystallization and phase transformation, the maximum flow stress of TC4 decreased with the increase in forming temperature.

The volume fraction of the β phase decreased with the increase in heating rate, as shown in Figure 9b. Specifically, when using a forming temperature of 900 °C, it was 47% at a heating rate of 1 °C/s and increased to 34% and 20% with the increasing heating rate of 10 °C/s and 100 °C/s, respectively. This was due to the fact that the phase transformation was determined by the diffusion of alloying elements. As the heating rate increased, the β phase stable element V lacked enough time to diffuse, resulting in an insufficient transformation of the β phase at this temperature. Consequently, the volume fraction of the β phase for the high heating rate condition was less than that of the equilibrium phase transformation process obtained from the isothermal condition [26]. Therefore, a higher heating rate applied in the hot deformation contributed to a higher maximum flow stress with less phase transformation.

In theory, the increase in forming temperature enhances the volume fraction of the phase, thus improving the elongation. However, severe grain coarsening at high temperatures may overshadow the decrease in elongation. At the same time, the increase in heating rate might lead to a decrease in the volume fraction of the α to β phase transformation, the combined effect of the two leading to a lower elongation during rapid heating at the forming temperatures of 850 °C and 900 °C.

The grain microstructure and size of TC4 were also measured at different heating rates based on the EBSD observations, as shown in Figure 10. It was found that the average grain size decreased from 10.16 μm at 10 °C/s to 8.76 μm at 100 °C/s, indicating that the rapid heating suppressed grain coarsening. The rate of the phase transformation was significant at 950 °C, and thus a large number of primary α transformed into β, significantly deteriorating the heating rate effect. In contrast, a slower heating rate caused the material to remain at an elevated temperature for a longer period, reducing the volume fraction of the α phase as well as the grain boundary movement hindrance of the β phase. Therefore, it was concluded that the grain size played a dominant role in the mechanical properties of TC4. The finer grains caused by rapid heating allowed the deformation to be dispersed in more grains, thus reducing the dislocation plugging in each grain and effectively improving the elongation at elevated temperatures.

### 3.3. Model of Phase Transformation at Different Heating Temperatures and Rates

For the dual-phase titanium alloys, the β phase with a bcc crystal structure is softer than the α phase with an hcp crystal structure. Hence, the volume fraction of the β phase has a large influence on the mechanical properties of the material. In addition, this parameter was found to be temperature-dependent based on the experimental observations. Consequently, the Johnson-Mehl-Avrami (JMA) equation describing the phase transformation from α to β under the isothermal conditions was used in the present research to model the relationship between the volume fraction of the β phase and the deformation temperature, as shown in Equation (1) [27]:(1)$$f_{\beta }={\left(\frac{T}{1270}\right)}^{10}$$
where *f_β_* was the volume fraction of the β phase, and *T* was the deformation temperature.

Considering the non-isothermal conditions, especially at a high heating rate, the phase transformation process is not equilibrated any longer. Therefore, the effect of the heating rate on the non-equilibrium phase transformation process was defined as Equation (2) [28]:(2)$$f_{T}=1-\exp\{-{\left[K(T-T_{0})/H\right]}^{n}\}$$
where *f_T_* was the temperature-dependent transformation volume fraction, n was the exponent of the transformation from α to β, T_0_ was the activation transformation temperature at the current heating rate, H was the heating rate, and K was a temperature-dependent constant, which was modelled by using the Arrhenius equation as below.
(3)$$K(T)=K_{0}exp(-\frac{Q}{RT})$$
where *K*_0_ was a constant and *Q* was the diffusion activation energy of the phase transformation from α to β.

The initial temperature of the phase transformation was affected by the heating rate. As a result, the phase transformation temperature demonstrated hysteretic behavior with the increase in the heating rate, as shown in Equation (4).
(4)$$T_{0}=T_{1}*[1+H/(a*H_{\max})]$$
where *T*_1_ was the activation temperature of the β phase equilibrium transformation, and *H*_max_ is the ultimate heating rate of Gleeble and is fixed at 1000 °C/s.

Consequently, the phase variation volume fraction of TC4 during continuous heating could be predicted by Equation (5), enabling the model to predict the volume fraction of the β phase at different temperatures and heating rates.
(5)$$f_{\beta }=f_{0}+f_{T}$$
where, *f*_0_ is the β phase volume fraction of TC4 titanium alloy at room temperature.

The experimental conditions of 850 °C–1 °C/s, 950 °C–1 °C/s, 900 °C–1 °C/s, 900 °C–10 °C/s and 900 °C–100 °C/s were selected as the orthogonal experiments, and each parameter in the equation was then calibrated using the Matlab fitting method as shown in Table 2. As shown in Figure 11, the experimental results of the volume fraction of the β phase were consistent with the modeling results with an average error of 5.17%, indicating a good accuracy of the proposed model. Based on the present research, the effect of cooling on the post-form strength and microstructure of TC4 will be used in the future to develop a theoretical model to predict the mechanical strength of formed components.

### 3.4. Effects of the Forming Temperature and Heating Rate on the Post-Form Strength of TC4

This study investigated the effects of forming temperature and heating rate on the strength of TC4 sheets that were heat treated using a self-developed resistance heating device. The strength properties at room temperature were characterized using the sub-size specimen shown in Figure 4b. For an easier comparison of the yield and ultimate tensile strength, engineering stress-strain curves were plotted in Figure 12, and the distribution of determined yield strength at various heating rates is shown in Figure 13.

It was clear that the heating rate affected the yield strength significantly, which decreased gradually with the increase in heating rate at the forming temperatures of 850 °C and 900 °C, but first increased and then decreased at the forming temperature of 800 °C. This trend was more obvious at a higher forming temperature. The difference in the yield strength between the heating rates of 30 °C/s and 100 °C/s was 16% at 800 °C, which was lower than that at 900 °C (24%). This was because when the heating rate was lower, the diffusion of alloying element V was more adequate, leading to a higher volume fraction of the β phase. It was known that the metastable β phase transformed to the fine and dislocated irregular acicular α’ phase, instead of the α phase, after cooling at a rate that was higher than the critical value of the martensitic phase transformation of TC4 [29]. Due to the α’ phase being stronger than the α phase, the more the β phase generated at a lower heating rate, the more it transformed into the α’ phase, resulting in a higher yield strength.

## 4. Conclusions

In this paper, the effects of heating rate and forming temperature on the plastic deformation, microstructure and post-forming properties of a dual-phase titanium alloy TC4 were systematically studied. The results of which would facilitate the precise control of the shape and mechanical properties of the formed titanium components under hot forming conditions. Therefore, the present research would contribute to the manufacture of complex-shaped thin-walled aerospace components made from dual-phase titanium alloys. The effects of the forming temperature and heating rate on the maximum flow stress and elongation of TC4 were investigated by conducting hot tensile tests and microstructural analysis. Subsequently, the yield strength at different heating rates was compared using the specimens that were heat treated using a resistance heating device. The main conclusions were drawn as below:

In the forming temperature range between 850 °C and 950 °C, both the maximum flow stress and elongation decreased with an increase in forming temperature. When the heating rate was 10 °C/s, the flow stress was larger than that at the heating rate of 1 °C/s, while the elongation remained constant.For the microstructure evolution under various heating conditions, the volume fraction of the β phase increased with an increase in heating temperature and a decrease in heating rate. The average grain size decreased with an increased heating rate. A higher volume fraction of the β phase and finer grains improved the material ductility.Based on the microstructure observation results, a model was established to predict the volume fraction of the β phase under different heat treatment conditions. The prediction error of the model was 5.17%, which would contribute to a qualitative analysis of the mechanical properties of TC4 titanium alloy under high-temperature deformation conditions.More martensite transformation was involved in the metastable β phase at a higher heating temperature and a lower heating rate during the rapid cooling process, leading to a higher yield strength.

## Figures and Tables

**Figure 1 materials-16-00810-f001:**
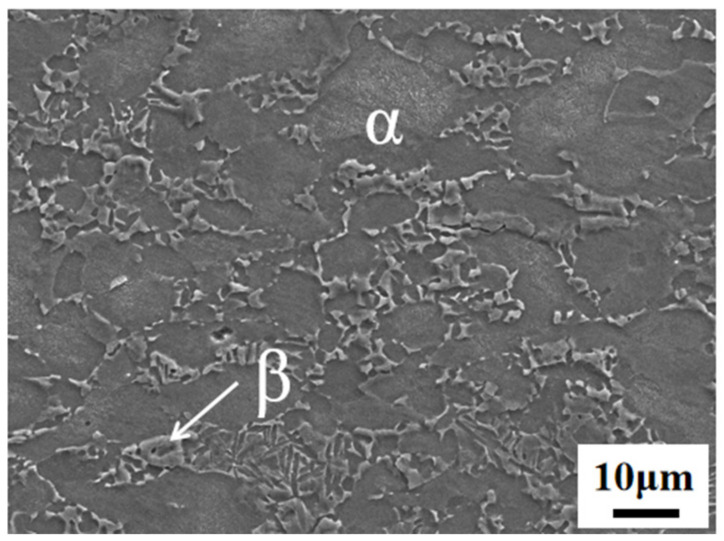
Microstructure of the as-received TC4 titanium alloy.

**Figure 2 materials-16-00810-f002:**
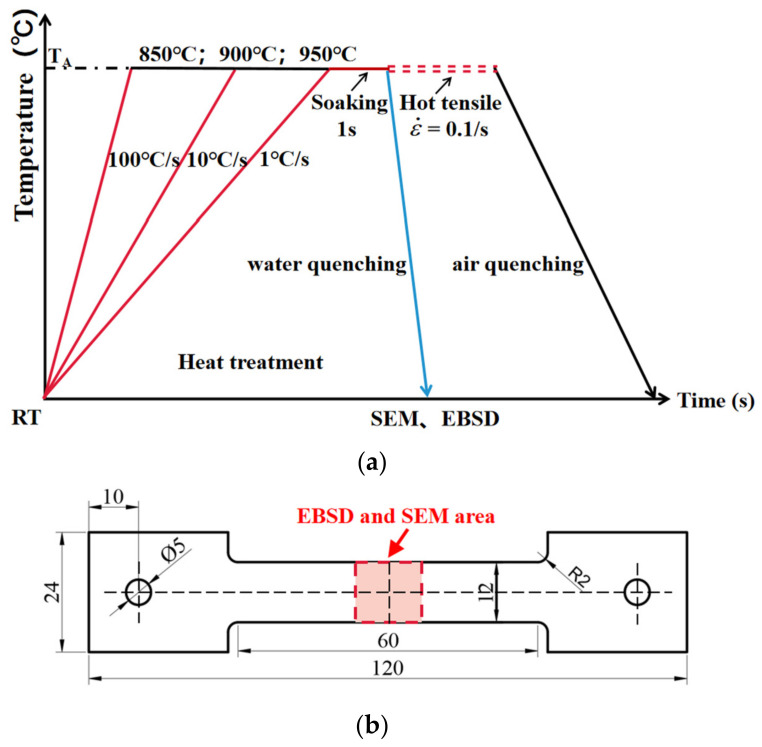
(**a**) Temperature profiles of the Gleeble hot tensile and heat treatment tests; (**b**) Geometries of Gleeble hot tensile tests and microstructure specimens.

**Figure 3 materials-16-00810-f003:**
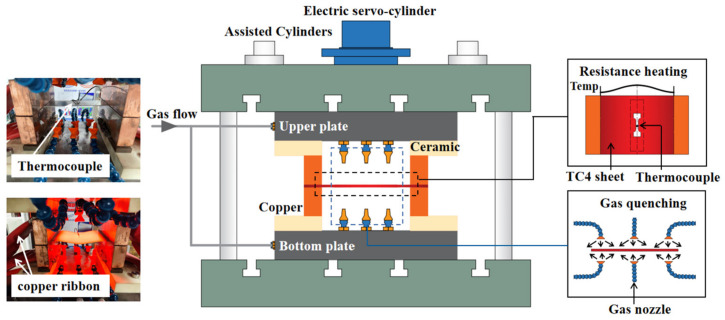
Schematic diagram of the resistance heat treatment device.

**Figure 4 materials-16-00810-f004:**
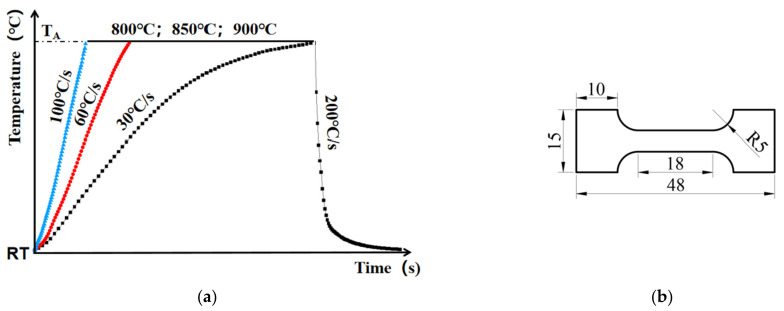
(**a**) Temperature histories of resistance heating experiments; (**b**) Geometry of the Gleeble tensile test specimens.

**Figure 5 materials-16-00810-f005:**
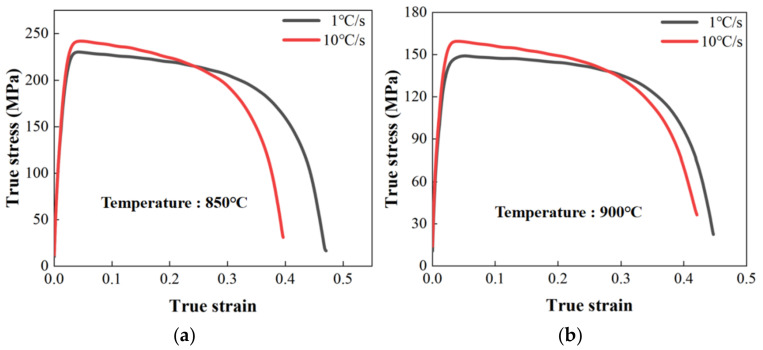
Effect of heating rate on the hot stress strain behaviors of TC4 at heating temperatures of (**a**) 850 °C; (**b**) 900 °C; (**c**) 950 °C.

**Figure 6 materials-16-00810-f006:**
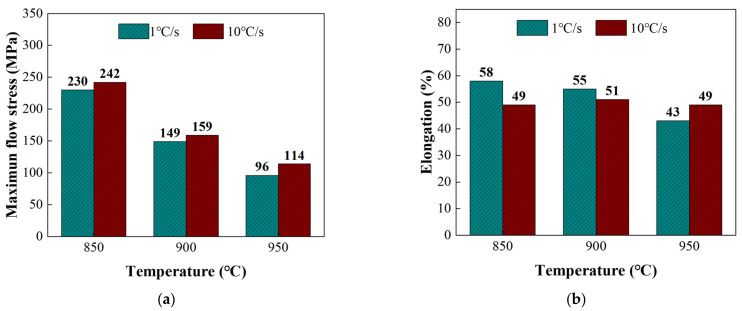
Effects of forming temperature and heating rate on (**a**) the maximum flow stress and (**b**) elongation.

**Figure 7 materials-16-00810-f007:**
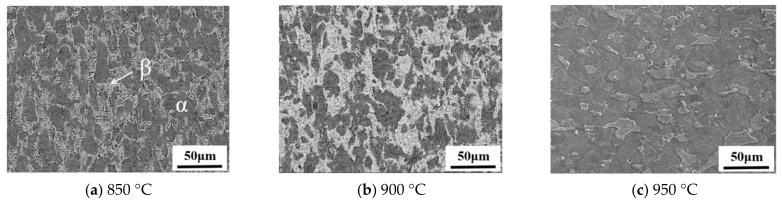
SEM observations of the α and β phases at different forming temperatures and a constant heating rate of 1 °C/s.

**Figure 8 materials-16-00810-f008:**

SEM observation of the α and β phases at different heating rates and a constant forming temperature of 900 °C.

**Figure 9 materials-16-00810-f009:**
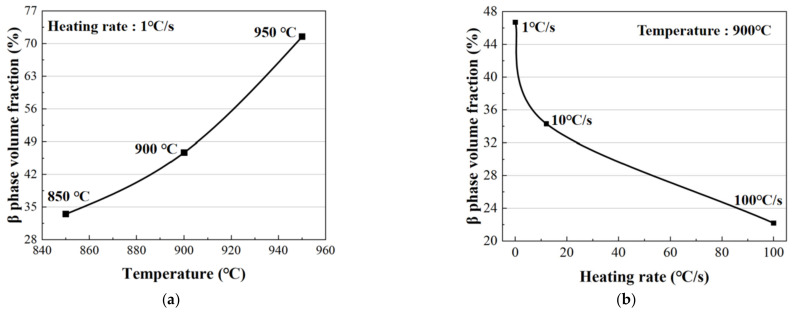
Volume fractions of the β phase at different (**a**) forming temperatures and (**b**) heating rates.

**Figure 10 materials-16-00810-f010:**
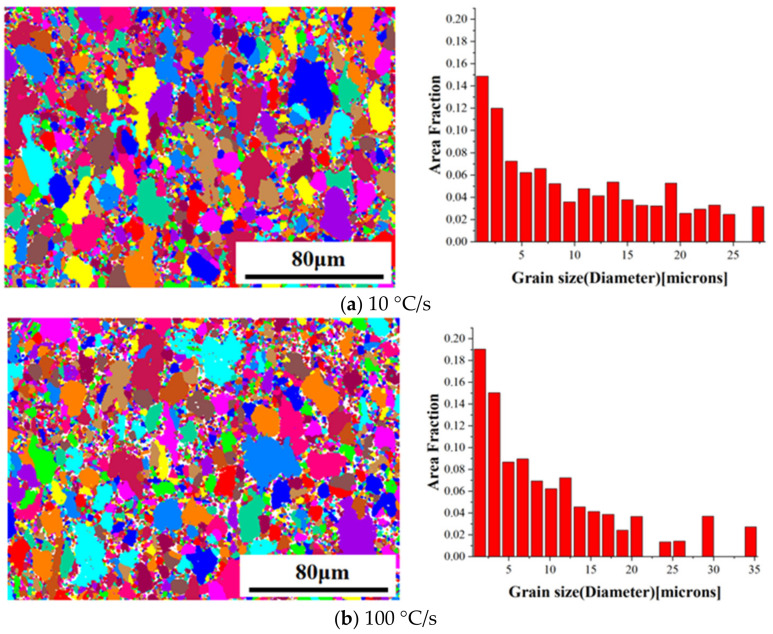
EBSD of the grain microstructure and size of TC4 at different heating rates and a constant forming temperature of 950 °C.

**Figure 11 materials-16-00810-f011:**
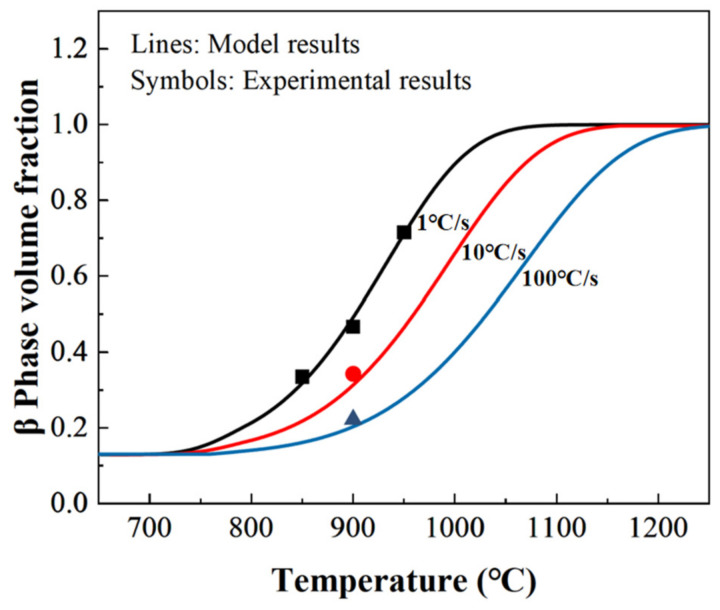
Comparisons of the experimental and modelling results on the volume fraction of the β phase at different heating rates and heating temperatures.

**Figure 12 materials-16-00810-f012:**
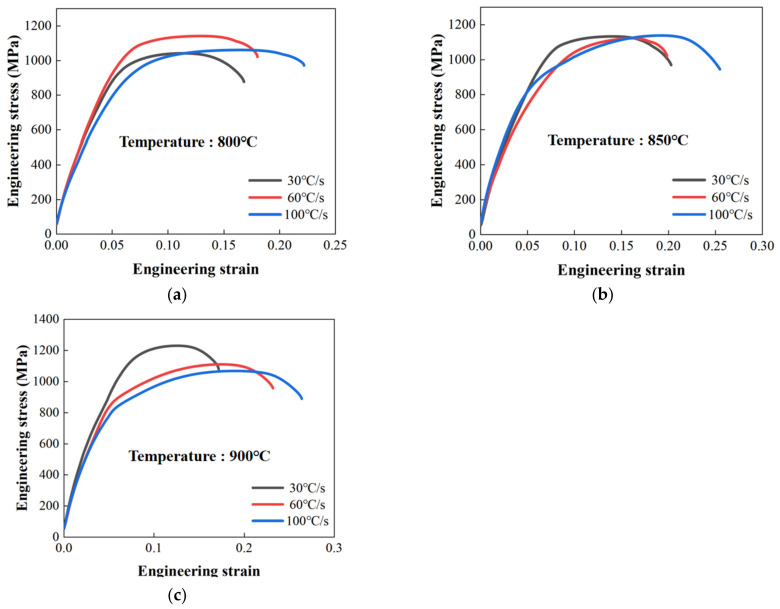
Effect of heating rate on the engineering stress-strain curves at the heating temperatures of (**a**) 800 °C (**b**) 850 °C; (**c**) 900 °C.

**Figure 13 materials-16-00810-f013:**
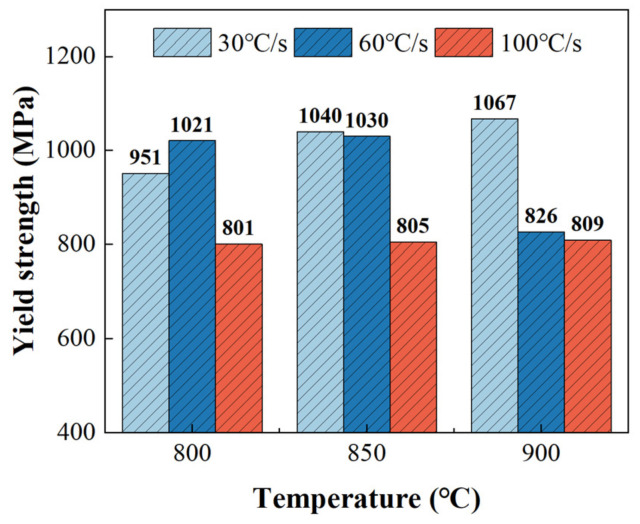
Yield strength after cooling at a cooling rate of 200 °C/s at different heating temperatures and rates.

**Table 1 materials-16-00810-t001:** Chemical composition of the as-received TC4 in weight percentage.

Ele.	Al	V	Fe	C	N	H	O	Ti
wt%	6.1	4.2	0.15	<0.01	<0.01	0.007	0.13	Bal.

**Table 2 materials-16-00810-t002:** Material constants of a phase transformation volume fraction prediction model.

Parameter	Value	Parameter	Value	Parameter	Value
*f* _0_	0.13	*K*_0_ (s^−1^)	4.52 × 10^16^	*Q* (kJ/mol)	445
*T*_1_ (K)	923	a	0.86	*H*_max_ (°C/s)	1000
n	0.35	R (J/mol/K)	8.314		

## Data Availability

Not applicable.

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
