# Peer review of "Effect of Heating on Hot Deformation and Microstructural Evolution of Ti-6Al-4V Titanium Alloy"

_materials, 2023, doi:10.3390/ma16020810_

Round 1

Reviewer 1 Report

Review

 The work is devoted to the study of the effect of heat treatment of titanium alloy Ti-6Al-4V on its microstructure and deformation properties. Standard tensile tests (Gleeble hot tensile tests) were carried out depending on the temperature and heating rate under normal conditions (temperature - 850 ° C, 900 ° C and 950 ° C, heating rate - 1 ºC / s and 10 ºC / s, strain rate 0.1/s). It has been found that the yield strength and ultimate strain of the Ti-6Al-4V titanium alloy decrease with an increase in the heat treatment temperature. The phase transformation of alpha titanium into β titanium was studied depending on the heat treatment temperature and heating rate. The volume fraction of β phase increased with increasing heating temperature and decreasing heating rate. The fitting method was used to determine the parameters of the model describing the transformation of alpha titanium into beta titanium depending on the heating conditions. Tensile tests were carried out on samples after heat treatment depending on the heat treatment temperature under normal conditions (temperature - 800°C, 850°C and 900°C, heating rate - 30ºC/s, 60ºC/s and 100ºC/s, cooling rate 200ºC/s). with). A martensitic transformation of the metastable β-phase upon rapid cooling was found, which leads to an increase in the yield strength. The decrease in the yield strength with an increase in the volume fraction of the beta phase of titanium has also been studied many times.

Remarks

1. The main drawback of the work is the limited theoretical description of the obtained results. The obtained experimental regularities are not new in themselves. The decrease in the yield strength of titanium alloy Ti-6Al-4V with an increase in its temperature durin heat treatment has been studied many times. The phenomenon of the average grain size decreasing with increasing heating rate has also been known for a long time. The decrease in the yield strength with an increase in the volume fraction of the beta phase of titanium has also been studied many times. At present, the theoretical description of these phenomena is of interest. The paper presents only a model for the transformation of alpha titanium into beta titanium. There are no models in the work that describe the plastic deformation and yield strength of a titanium alloy during heat treatment and after heat treatment, depending on the heat treatment conditions on the phase composition. It is necessary to work with a theoretical model that describes the effect of heat treatment conditions on the hot deformation of a titanium alloy. It is possible to give a theoretical model for the data presented in fig.6 and fig.13.

2. The used model has 8 parameters (table 2). These eight values were fitted to six experimental points. It is impossible to make a correct fit of 8 parameters using 6 points. With this approach, 2 degrees of freedom remain, allowing arbitrary choice of parameters. It is necessary to provide an explanation, increase the number of experiments, use parameter values from other sources, or use a simpler model.

3. The unit of measurement s-1 of parameter K0 in table 2 is presented incorrectly. Probably meant s-1.

Reviewer 2 Report

1.     Research novelty should be highlighted at the end of literature review.

2.     Latest literature review on Ti-6AL-4V should be cited.

Surya, M.S., Prasanthi, G., Kumar, A.K. et al. Optimization of cutting parameters while turning Ti-6Al-4 V using response surface methodology and machine learning technique. Int J Interact Des Manuf 15, 453–462 (2021). https://doi.org/10.1007/s12008-021-00774-0

Mulugundam Siva Surya, Optimization of turning parameters while turning Ti-6Al-4V titanium alloy for surface roughness and material removal rate using response surface methodology, Materials Today: Proceedings, Volume 62, Part 6, 2022, Pages 3479-3484, ISSN 2214-7853, https://doi.org/10.1016/j.matpr.2022.04.300.

Reviewer 3 Report

materials-2124660-peer-review-v1                 Review Report

Title of the Paper : Effect of heating on hot deformation and microstructural evolu-

tion of Ti-6Al-4V titanium alloy

The work is good and executed well. However following suggestions/corrections needs to be incorporated for publication.

1.      The results and discussion header should appear on page 6.

2.      Need more explanation about Figs 7 and 8.

3.      Why higher yield strength observed at a lower heating rate?

4.      Explain what the maximum flow stress and yield strength would be if the heating rate exceeded 10%.

5.      Please check the true stress and true strain curves in Fig. 5.

6.      must include current articles in references

Round 2

Reviewer 1 Report

The authors added explanations and eliminated major remarks. Not all results have a theoretical explanation. However, due to the great technical significance, the article can be recommended for publication.

Reviewer 2 Report

Accepted